# The correlation between NAFLD and serum uric acid to serum creatinine ratio

**Jangwon Choi** [1], **Hyun Joe** [1]*, **Jung-Eun Oh** [2], **Yong-Jin Cho** [2], **Hwang-Sik Shin** [2], **Nam Hun Heo** [3]

**1** Department of Family Medicine, College of Medicine, Soonchunhyang University College of Medicine, Soonchunhyang University Seoul Hospital, Seoul, Korea, **2** Department of Family Medicine, College of Medicine, Soonchunhyang University Cheonan Hospital, Chungnam, Republic of Korea, **3** Biostatics Department of Clinical Trial Center, College of Medicine, Soonchunhyang University Cheonan Hospital, Chungnam, Korea

☯ These authors contributed equally to this work.

\* drjoe@schmc.ac.kr

**Data Availability Statement:** All relevant data are within the paper.

**Funding:** The authors received no specific funding for this work.

## Abstract

### Background

With the prevalence of non-alcoholic fatty liver disease (NAFLD) increasing worldwide, many noninvasive techniques have been used to improve its diagnosis. Recently, the serum uric acid/creatinine (sUA/sCr) ratio was identified as an indicator of fatty liver disease. Therefore, we examined the relationship between sUA/sCr levels and ultrasound-diagnosed NAFLD in Korean adults.

### Methods

This study included 16,666 20-year-olds or older who received health checkups at a university hospital's health promotion center from January to December 2021. Among them, 11,791 non-patients with and without NAFLD were analyzed, excluding those without abdominal ultrasound, those without data on fatty liver, cancer, or chronic kidney disease severity, those with a history of alcohol abuse, and those with serum hs-CRP <5 mg/L. The odds ratio (OR) and 95% confidence interval (CI) of the sUA/sCr ratio according to the presence or absence of fatty liver disease and severity were calculated after correcting for confounding variables using logistic regression analysis. The receiver operating characteristic (ROC) curve and area under the curve (AUC) of the sUA/sCr ratio confirmed and compared the sensitivity and specificity of NAFLD and serum uric acid.

### Results

sUA/sCr increased with fatty liver severity, and the post-correction OR in the NAFLD group was 1.183 (95% CI: 1.137–1.231) compared to the group without NAFLD. Concerning the fatty liver severity, the post-correction OR in the mild NAFLD group increased to 1.147 (95% CI: 1.099–1.196), and that in the moderate-to-severe NAFLD group increased to 1.275 (95% CI: 1.212–1.341) compared to the group without NAFLD. The sensitivity of sUA/sCr to fatty liver severity was 57.9% for the non-NAFLD group, 56.7% for the mild NAFLD group, and 59.0% for the moderate-to-severe NAFLD group; the specificity of sUA/sCr to fatty liver

**Competing interests:** The authors have declared that no competing interests exist.

severity 61.4% for the non-NAFLD group, 57.3% for the mild NAFLD group, and 65.2% for the moderate-to-severe NAFLD group.

## Conclusion

NAFLD severity is associated with sUA/sCR.

## Introduction

Non-alcoholic fatty liver disease (NAFLD) is characterized by the accumulation of more than 5% fat in the liver without a history of alcohol or drugs [1] and is an intrahepatic finding of the metabolic syndrome [2]. MetS increases the risk of developing diabetes, dyslipidemia, arteriosclerosis, and various cardiovascular and cerebrovascular diseases [3]. Recently, the prevalence of NAFLD has increased worldwide with an increase in the number of patients diagnosed with metabolic syndrome [4]. NAFLD has emerged as a major health problem, leading to an increase in total medical costs, socioeconomic burden, and individual health problems owing to its serious complications [5, 6].

NAFLD includes various types of liver diseases, ranging from non-alcoholic fatty liver without liver cell damage and fibrosis to chronic hepatitis and cirrhosis, which are characterized by fat deposition in the liver [7]. Fatty liver can be classified based on the degree of local fat deposition as mild, moderate, or severe [8]. Fat hepatitis is a condition characterized by inflammation accompanied by hepatocellular damage in patients with fat deposition in the liver and may be accompanied by fibrosis [7].

The prevalence of hyperuricemia and gout is also increasing, owing to changes in eating habits and lifestyles and an increase in the population of older adults [9]. According to recent studies, serum uric acid is the final metabolite of purines, and elevated serum uric acid levels can contribute to insulin resistance through various mechanisms [10–12], ultimately leading to metabolic syndrome and NAFLD, according to the results of recent studies [13–17]. As uric acid is excreted through the kidneys, blood uric acid levels are affected by kidney function. Therefore, there is a possibility that blood uric acid/creatinine regarding kidney function may be a more useful parameter than serum uric acid levels alone. Several studies have confirmed the correlation between NAFLD and the serum uric acid/creatinine ratio (sUA/sCr) [18–22]. However, to date, no studies have examined the relationship between NAFLD identified by ultrasonography, which is mainly used for fatty liver diagnosis, and the sUA/sCr ratio in Koreans.

Currently, there is no established screening method for NAFLD; however, NAFLD is mainly diagnosed using abdominal ultrasonography, and liver biopsy has been suggested as a confirmatory method [7]. Because this method is invasive, diagnosis using radiology has been preferred, and there have been many attempts to increase the diagnosis rate using noninvasive methods with clinical and biochemical indicators [23]. This study aimed to examine the relationship between NAFLD diagnosed using ultrasound and sUA/sCr in healthy Koreans visiting a screening center and to assess the usefulness of sUA/sCr as a new indicator for diagnosing NAFLD.

## Materials and methods

### Participants

From January 2021 to December 2021, 16,666 examinees aged 20 or older who received medical checkups at the Health Examination Center of Soonchunhyang University Cheonan

Hospital in Korea were surveyed. In this study, NAFLD was diagnosed in those with a fatty liver but with alcohol intake less than 210 g/week for men and 140 g/week for women, with no drug use indicative of viral hepatitis history or fat accumulation in the liver, and negative screening results for hepatitis B virus antigen and hepatitis C virus antibody [24].

A total of 1,453 individuals were excluded because of a lack of ultrasound examination records or fatty liver severity confirmed by ultrasound. A further 1,064 people were excluded because they were HBsAg positive or had anti-HCV antibodies (based on blood test results) and responded as having HBV(Hepatitis B virus) and/or cancer on the questionnaires. Additionally, 1,967 people whose alcohol intake exceeded 210 g/week for men and 140 g/week for women were excluded to exclude cases of alcoholic fatty liver disease. A total of 42 people who responded that they had chronic kidney disease on the questionnaire or whose eGFR was <60 mg/dL (based on blood tests) were excluded. To exclude the possibility of other diseases such as acute infection, 324 patients with serum hs-CRP >5 mg/L and 24 patients with missing values were also excluded. The final study included 11,791 participants: 6,062 in the control group (without fatty liver disease) and 5,729 in the NAFLD group (**Fig 1**).

## Variables

The following information was obtained using a self-administered questionnaire: drug use, drinking history (number of alcoholic drinks per week, type of alcohol consumed (soju, beer, makgeolli, wine, and liquor), and amount of alcohol consumed in one sitting), and smoking history. The amount of alcohol consumed per week was calculated using the following formula: [the amount of alcohol (mL) × alcohol content (%) × alcohol specific gravity (0.79)/100 = amount of alcohol contained in alcohol (g)] [25]. Participants were classified as nonsmokers, past smokers, or current smokers based on their smoking history. The exercise was calculated by converting it into a Metabolic Equivalent Task (MET [minutes]) according to the Korean version of the Global Physical Activity Questionnaire (K-GPAQ) scoring system [26] and classified into two groups: less than 600METs/week in the low physical activity group and more than 600 METs/week in the physically active group [27]. Body mass index (BMI) was calculated by dividing the weight in kilograms (measured using an electronic body composition analyzer (X-Scan plus 2, Jawon Medical Co. Ltd., Korea)) by the square of the height (in meters squared); participants were classified as normal if their BMI was between 18 and 25 kg/m$^2$, according to the obesity standard of Koreans [28, 29]. Blood pressure was the mean of two measurements in the upper left arm using an EASY X 900 instrument (Jawon Medical Co., Ltd., Korea; Colin Electronics Co., Ltd., Aichi, Japan); if it was 140/90 mmHg or more, a manual blood pressure meter (DS44-11; Welch Allen, Salkane, New York, USA) was used for reassessment and considered as the final value. Waist circumference was measured directly with a tape measure by measuring the middle area between the lowest rib and pelvic iliac region with the participant in an upright position, breathing lightly, according to the World Health Organization guidelines [30], and recording up to 0.1 cm. According to the Korean abdominal obesity standards [31], men were considered obese if their waist circumference was 90 cm or greater; for women, the cutoff value was 85 cm. After fasting for more than 10 hours, peripheral venous blood samples were collected for assessment of fasting blood sugar (FBS), glycated hemoglobin (HbA1C), low-density lipoprotein cholesterol (LDL-C), high-density lipoprotein cholesterol (HDL-C), total cholesterol, triglyceride (TG), aspartate aminotransferase (AST), alanine aminotransferase (ALT), γ-glutamyl transferase (γ-GT), serum creatinine, serum uric acid, and hs-CRP levels. Upper abdominal ultrasound was performed using a Philips iU22 (Philips, Amsterdam, Netherlands), and the findings were read by two experienced radiologists to confirm the diagnosis of fatty liver [32] (which was performed when the echo of the hepatic parenchyma was higher than that of the right kidney medulla).

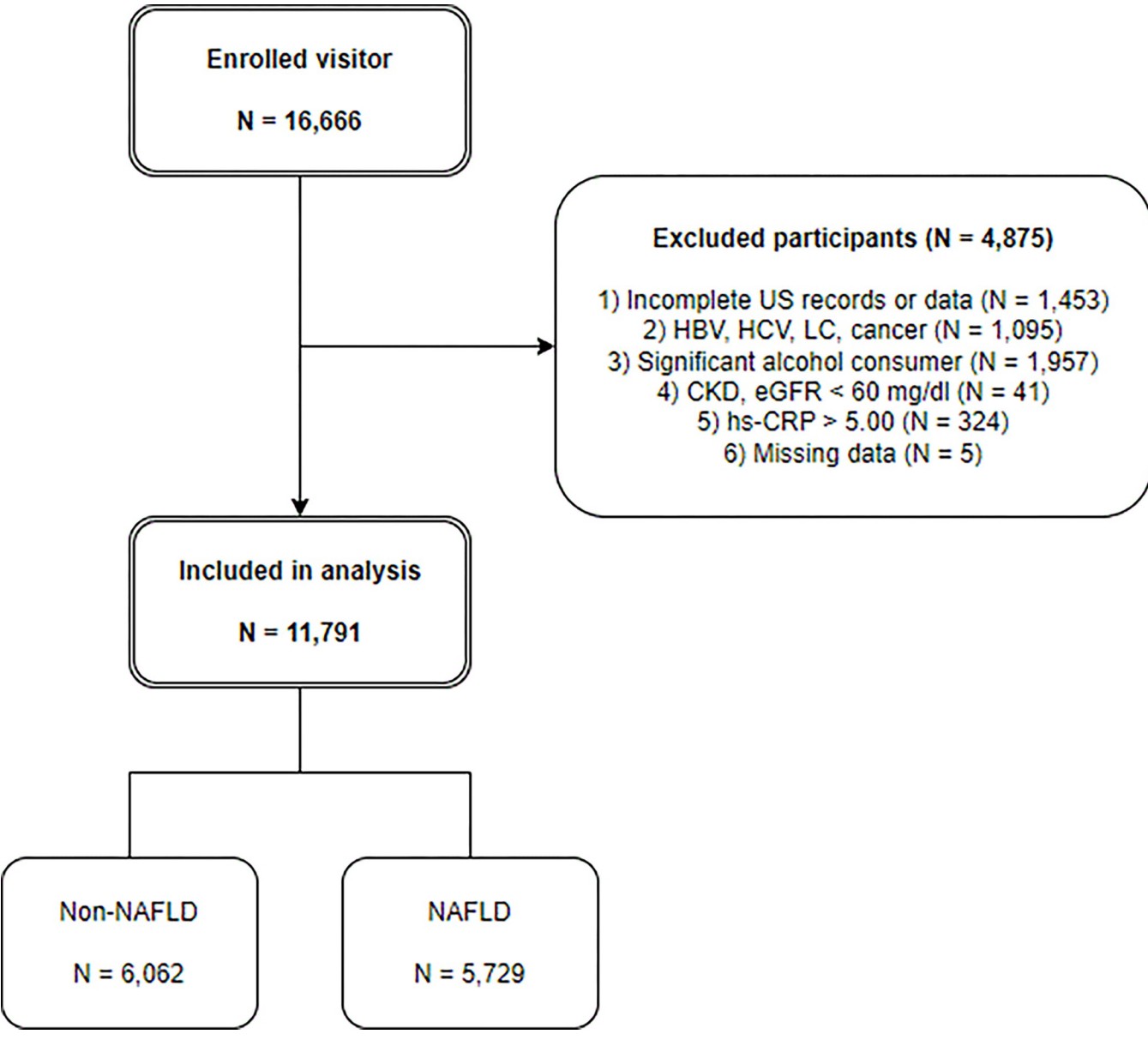

**Fig 1. Study flow diagram.** US; Ultrasonography, HBsAG; Hepatitis B virus surface antigen, LC; Liver cirrhosis, CKD; chronic kidney disease.

## Statistical analysis

All statistical analyses were performed using IBM SPSS Statistics ver. 26.0 (IBM Corp., Armonk, NY, USA). Normality tests were conducted on continuous variables using both the Kolmogorov-Smirnov test and the Shapiro-Wilk test. Student's t-test, or one-way analysis of variance, was used to analyze continuous variables, whereas Pearson's chi-square test was used to analyze categorical variables.

The participants were classified into the following three groups according to the severity of fatty liver disease on ultrasonography: normal (non-NALFD), mild (mild NAFLD), and moderate-to-severe fatty liver and cirrhosis groups (moderate-to-severe NAFLD).

The Odds ratio (OR) and 95% confidence interval (CI) of the sUA/sCr ratio according to the presence or absence of fatty liver disease and according to the severity of fatty liver disease

were calculated after adjusting for confounding variables using logistic regression analysis. The sensitivity and specificity of the sUA/sCr ratio for NAFLD were confirmed using the receiver operating characteristic (ROC) curve and area under the curve (AUC). The optimum cutoff value was at which the sum of the sensitivity and specificity was maximal.

In this study, a *p*-value of <0.05 was considered significant.

### Ethics statement

Research participants provided written informed consent at the time of visiting. Participants were informed that they could withdraw from the study at any time. This study protocol was reviewed and approved by the Institutional Review Board of Soonchunhyang University Cheonan Hospital on March 29th, 2023 (IRB Number: SCHCA 2023-03-021). All procedures, including data collection and analysis, were conducted after March 29th, 2023 in accordance with the Personal Information Protection Act of Korea.

## Results

### Characteristics of the study population

The results of the comparison based only on the presence or absence of NAFLD are presented in Table 1, and those based on the results according to the severity of NAFLD are presented in Table 2. The average age of the entire study cohort was 44.67 ± 9.61 years; it was 43.81 ± 9.88 years in the non-NAFLD group, 46.20 ± 9.67 years in the mild NAFLD group, and 45.02 ± 8.80 years in the moderate-to-severe NAFLD group. Of the 11,791 study participants, 6,847 (58.07%) were men and 4,944 (41.93%) were women, and the number of male participants increased with the severity of NAFLD. Regarding smoking status, 6,454 (57.24%) were nonsmokers, 2,585 (22.92%) were past smokers, and 2,237 (19.84%) were current smokers. Furthermore, 5,251 (44.54%) participants were in the low physical activity group and 6,539 (55.46%) were in the high physical activity group in this study.

The average BMI was 24.45 ± 3.61 kg/m$^2$, and 4,770 (40.45%) individuals were obese. The average BMI of the non-NAFLD group was 22.50 ± 2.61 kg/m$^2$, and 987 individuals (16.27%) were obese; in the mild NAFLD group, the BMI was 25.25 ± 2.71 kg/m$^2$, and 1,376 (51.00%) individuals were obese. In the moderate-to-severe NAFLD group, BMI was 27.64 ± 3.47 kg/m$^2$, and 2,407 (79.41%) were obese. The proportion of obese individuals increased with the severity of fatty liver disease.

The average systolic blood pressure of all individuals was 117.48 ± 11.60 mmHg, and the average diastolic blood pressure was 72.10 ± 9.28 mmHg. The average blood pressure increased with the increasing severity of the fatty liver disease.

The average waist circumference was 82.59 ± 10.41 cm, and 3,220 (27.34%) had abdominal obesity. In the non-NAFLD group, the average waist circumference was 76.56 ± 8.05 cm, and the abdominal obesity value was 468 (7.74%); in the mild NAFLD group, the average waist circumference was 85.39 ± 7.24 cm, abdominal obesity 844 (31.28%); in the moderate-to-severe NAFLD group, average waist circumference was 92.16 ± 8.60 cm, and abdominal obesity was 1,908 (62.95%). As the severity of fatty liver disease increased, waist circumference and abdominal obesity increased proportionally.

AST/ALT, γ-GTP, FBS, HbA1C, total cholesterol, TG, LDL-C, HDL-C, and hs-CRP all increased with an increase in NAFLD severity (lowest in the non-NAFLD group and highest in the moderate-to-severe NAFLD group).

The sUA level was 5.56 ± 1.50 mg/dL on average overall, and 2,102 patients (17.92%) had hyperuricemia (7 mg/dL or more). In the non-NAFLD group, of serum uric acid (UA) level was 5.03 ± 1.32 mg/dL, and 507 (8.41%) participants had hyperuricemia; the average UA level

**Table 1. Characteristics of the study population (non-NAFLD vs NAFLD).**

| | Total | Non-NAFLD | NAFLD | *p*-value |
|---|---|---|---|---|
| **No. of subjects** | 11,791 | 6,062 | 5,729 | |
| **Age** | 44.67 ± 9.61 | 43.81 ± 9.88 | 45.58 ± 9.24 | <0.001 |
| **Sex** | | | | |
| Male | 6,847 (58.07) | 2,547 (42.02) | 4,300 (75.06) | <0.001 |
| Female | 4,944 (41.93) | 3,515 (57.98) | 1,429 (24.94) | |
| **Smoking** | | | | |
| Non-smoker | 6,454 (57.24) | 4,011 (69.20) | 2,443 (44.58) | <0.001 |
| Ex-Smoker | 2,585 (22.92) | 970 (16.74) | 1,615 (29.47) | |
| Current-Smoker | 2,237 (19.84) | 815 (14.06) | 1,422 (25.95) | |
| **Exercise(METs/week)** | 694.01 ± 600.88 | 712.03 ± 618.72 | 674.94 ± 580.86 | 0.001 |
| <600 | 5,251 (44.54) | 2,673 (44.09) | 2,578 (45.01) | 0.319 |
| ≥600 | 6,539 (55.46) | 3,389 (55.91) | 3,150 (54.99) | |
| **BMI(kg/㎡)** | 24.45 ± 3.61 | 22.50 ± 2.61 | 26.51 ± 3.35 | |
| <25 | 7,021 (59.55) | 5,075 (83.72) | 1,946 (33.97) | <0.001 |
| ≥25 | 4,770 (40.45) | 987 (16.28) | 3,783 (66.03) | |
| **Systolic BP(mmHg)** | 117.48 ± 11.60 | 114.05 ± 11.62 | 121.12 ± 10.40 | <0.001 |
| <140 | 11,340 (96.18) | 5,935 (97.90) | 5,405 (94.34) | <0.001 |
| ≥140 | 451 (3.82) | 127 (2.10) | 324 (5.66) | |
| **Diastolic BP(mmHg)** | 72.10 ± 9.28 | 69.34 ± 9.01 | 75.02 ± 8.65 | <0.001 |
| <90 | 11,230 (95.24) | 5,913 (97.54) | 5,317 (92.81) | <0.001 |
| ≥90 | 561 (4.76) | 149 (2.46) | 412 (7.19) | |
| **Waist circumference(cm)** | 82.59 ± 10.41 | 76.56 ± 8.05 | 88.97 ± 8.67 | <0.001 |
| M<90, F<85 | 8,558 (72.66) | 5,581 (92.26) | 2,977 (51.96) | <0.001 |
| M≥90, F≥85 | 3,220 (27.34) | 468 (7.74) | 2,752 (48.04) | |
| **AST(IU/l)** | 24.51 ± 15.18 | 21.42 ± 13.21 | 27.78 ± 16.41 | <0.001 |
| **ALT(IU/l)** | 27.43 ± 23.02 | 19.05 ± 13.69 | 36.30 ± 27.20 | <0.001 |
| **γ-GTP(IU/l)** | 30.83 ± 37.24 | 21.47 ± 27.13 | 40.73 ± 43.41 | <0.001 |
| **FBS(mg/dl)** | 100.08 ± 18.54 | 95.01 ± 12.89 | 105.44 ± 21.81 | <0.001 |
| **HbA1c(%)** | 5.62 ± 0.64 | 5.44 ± 0.41 | 5.80 ± 0.78 | <0.001 |
| <6.5 | 10,968 (93.76) | 5,899 (98.09) | 5,069 (89.18) | <0.001 |
| ≥6.5 | 730 (6.24) | 115 (1.91) | 615 (10.82) | |
| **Total cholesterol(mg/dl)** | 205.78 ± 39.22 | 203.42 ± 36.63 | 208.27 ± 41.65 | <0.001 |
| **Triglyceride(mg/dl)** | 134.09 ± 91.38 | 100.76 ± 55.36 | 169.36 ± 107.35 | <0.001 |
| **LDL-C(mg/dl)** | 125.06 ± 34.40 | 121.63 ± 32.40 | 128.69 ± 36.05 | <0.001 |
| **HDL-C(mg/dl)** | 56.50 ± 15.09 | 62.86 ± 15.12 | 49.77 ± 11.77 | <0.001 |
| **ALP(IU/l)** | 72.76 ± 20.98 | 68.13 ± 20.41 | 77.66 ± 20.45 | <0.001 |
| **Total bilirubin** | 0.86 ± 0.41 | 0.84 ± 0.41 | 0.88 ± 0.42 | <0.001 |
| **Direct bilirubin** | 0.31 ± 0.12 | 0.31 ± 0.12 | 0.31 ± 0.12 | 0.586 |
| **Amylase(IU/l)** | 64.19 ± 22.90 | 68.23 ± 23.45 | 59.92 ± 21.49 | <0.001 |
| <110 | 11,168 (96.68) | 5,656 (95.35) | 5,512 (98.08) | <0.001 |
| ≥110 | 384 (3.32) | 276 (4.65) | 108 (1.92) | |
| **Lipase(U/l)** | 31.02 ± 14.35 | 30.61 ± 15.71 | 31.46 ± 12.73 | 0.004 |
| 60 | 9,529 (97.57) | 4,914 (97.97) | 4,615 (97.16) | 0.009 |
| ≥60 | 237 (2.43) | 102 (2.03) | 135 (2.84) | |
| **AFP** | 3.34 ± 6.03 | 3.54 ± 8.16 | 3.13 ± 2.08 | <0.001 |
| **hs-CRP(mg/l)** | 0.75 ± 0.83 | 0.52 ± 0.65 | 0.98 ± 0.93 | <0.001 |
| **Uric acid(mg/dl)** | 5.56 ± 1.50 | 5.03 ± 1.32 | 6.11 ± 1.48 | <0.001 |

*(Continued)*

**Table 1.** (Continued)

| | Total | Non-NAFLD | NAFLD | *p*-value |
|---|---|---|---|---|
| <7 | 9,626 (82.08) | 5,518 (91.59) | 4,108 (72.03) | <0.001 |
| ≥7 | 2,102 (17.92) | 507 (8.41) | 1,595 (27.97) | |
| **BUN(mg/dl)** | 12.60 ± 3.39 | 12.25 ± 3.39 | 12.97 ± 3.36 | <0.001 |
| **Creatinine(mg/dl)** | 0.85 ± 0.18 | 0.81 ± 0.18 | 0.89 ± 0.18 | <0.001 |
| **sUA/sCr** | 6.64 ± 1.55 | 6.34 ± 1.43 | 6.95 ± 1.61 | <0.001 |

Values are presented as means ±SD or number (%).

NAFLD: non-alcoholic fatty liver disease, BMI: body mass index, BP: blood pressure, FBS; Fasting glucose, HbA1C; Hemoglobin A1C, LDL-C: low-density lipoprotein-cholesterol, HDL-C: high-density lipoprotein-cholesterol, AST: aspartate aminotransferase, ALT: alanine aminotransferase, r-GTP: gamma-glutamyl transferase, ALP; Alkaline phosphatase.

was 5.78 ± 1.42 mg/dL in the mild NAFLD group, and 554 (20.60%) had hyperuricemia, while the average UA level was 6.41 ± 1.48% and 1,041 (34.54%) had hyperuricemia in the moderate-to-severe NAFLD group. Thus, with the increasing severity of fatty liver disease, sUA levels and the incidence of hyperuricemia increased.

The mean sCr level in the entire study cohort was 0.85 ± 0.18 mg/dL; it was 0.81 ± 0.18, 0.88 ± 0.18, and 0.91 ± 0.17 mg/dL in the non-NAFLD, mild NAFLD, and moderate-to-severe NAFLD groups, respectively. sCr levels increased with an increase in fatty liver severity.

The mean sUA/sCr ratio was 6.64 ± 1.55 in the entire cohort; this significantly increased from 6.34 ± 1.43 in the non-NAFLD group to 6.71 ± 1.49 in the mild NAFLD group and 7.17 ± 1.68 in the moderate-to-severe NAFLD group (*p*-value of <0.001).

## OR of sUA/sCr according to the presence or absence of fatty liver disease and as per the severity, sensitivity, and specificity

After adjusting for other variables, when the sUA/sCr ratio was 1 in the fatty liver-free group, the OR of the fatty liver group was 1.183 (95% CI: 1.137–1.231, $p < 0.001$). When the OR of the non-NAFLD group was viewed as 1, the OR of the mild NAFLD group increased significantly to 1.147 (95% CI: 1.099–1.196, $p < 0.001$) and that of the moderate-to-severe NAFLD group increased to 1.275 (95% CI: 1.212–1.341, $p < 0.001$) (**Tables 3** and **4**).

The AUC of the sUA/sCr ratio according to the severity of fatty liver was 0.619 (95% CI: 0.610–0.629), with and without fatty liver. It was 0.581 (95% CI: 0.566–0.595) in the mild NAFLD group and 0.653 (95% CI: 0.642–0.666) in the moderate-to-severe NAFLD group. This was lower than the 0.710 (95% CI: 0.704–0.719), 0.655 (95% CI: 0.642–0.667), and 0.760 (95% CI: 0.751–0.769) seen for each of the groups for sUA levels. In addition, the sensitivity and specificity of the sUA/sCr ratio for the presence or absence of fatty liver were 57.9% and 61.4% ($p < 0.001$), respectively; 56.7% and 57.3% ($p < 0.001$) for mild NAFLD; and 59.0% and 65.2% ($p < 0.001$) for moderate-to-severe NAFLD, respectively (**Table 5** and **Fig 2**).

The sensitivity and specificity of UA levels for the presence or absence of fatty liver were 70.9% and 60.6% ($p < 0.001$) overall, 62.7% and 60.6% ($p < 0.001$) for mild NAFLD, and 76.0% and 63.2% ($p < 0.001$) for moderate-to-severe NAFLD. Thus, the sensitivity and specificity of the sUA/sCr ratio were lower than those of sUA (**Table 6** and **Fig 3**).

## Discussion

In this study, the correlation between NAFLD and sUA/sCR in adults without a significant history of alcohol use was investigated, and sensitivity and specificity were compared with sUA

**Table 2. Characteristics of the study population according to NAFLD severity.**

| | Total | Non-NAFLD | Mild NAFLD | Moderate-to-Severe NAFLD | *p*-value | Post hoc test | Trend *p*-value |
|---|---|---|---|---|---|---|---|
| **No. of subjects** | 11,791 | 6,062 | 2,698 | 3031.000 | | | |
| **Age** | 44.67 ± 9.61 | 43.81 ± 9.88 | 46.20 ± 9.67 | 45.02 ± 8.80 | <0.001 | 0 < 1 < 2 | <0.001 |
| **Sex** | | | | | | | |
| Male | 6,847 (58.07) | 2,547 (42.02) | 1,824 (67.61) | 2,476 (81.69) | <0.001 | | <0.001 |
| Female | 4,944 (41.93) | 3,515 (57.98) | 874 (32.39) | 555 (18.31) | | | |
| **Smoking** | | | | | | | |
| Non-smoker | 6,454 (57.24) | 4,011 (69.20) | 1,279 (50.02) | 1,164 (39.82) | <0.001 | | <0.001 |
| Ex-Smoker | 2,585 (22.92) | 970 (16.74) | 697 (27.26) | 918 (31.41) | | | |
| Current-Smoker | 2,237 (19.84) | 815 (14.06) | 581 (22.72) | 841 (28.77) | | | |
| **Exercise(METs/week)** | 694.01 ± 600.88 | 712.03 ± 618.72 | 701.35 ± 601.00 | 651.44 ± 561.39 | <0.001 | 0 = 1 < 2 | <0.001 |
| <600 | 5,251 (44.54) | 2,673 (44.09) | 1,159 (42.97) | 1,419 (46.82) | 0.009 | | 0.034 |
| ≥600 | 6,539 (55.46) | 3,389 (55.91) | 1,538 (57.03) | 1,612 (53.18) | | | |
| **BMI(kg/㎡)** | 24.45 ± 3.61 | 22.50 ± 2.61 | 25.25 ± 2.71 | 27.64 ± 3.47 | <0.001 | 0 < 1 < 2 | <0.001 |
| <25 | 7,021 (59.55) | 5,075 (83.72) | 1,322 (49.00) | 624 (20.59) | <0.001 | | <0.001 |
| ≥25 | 4,770 (40.45) | 987 (16.28) | 1,376 (51.00) | 2,407 (79.41) | | | |
| **Systolic BP(mmHg)** | 117.48 ± 11.60 | 114.05 ± 11.62 | 119.65 ± 10.90 | 122.42 ± 9.76 | <0.001 | 0 < 1 < 2 | <0.001 |
| <140 | 11,340 (96.18) | 5,935 (97.90) | 2,570 (95.26) | 2,835 (93.53) | <0.001 | | <0.001 |
| ≥140 | 451 (3.82) | 127 (2.10) | 128 (4.74) | 196 (6.47) | | | |
| **Diastolic BP(mmHg)** | 72.10 ± 9.28 | 69.34 ± 9.01 | 73.64 ± 8.78 | 76.25 ± 8.34 | <0.001 | 0 < 1 < 2 | <0.001 |
| <90 | 11,230 (95.24) | 5,913 (97.54) | 2,551 (94.55) | 2,766 (91.26) | <0.001 | | <0.001 |
| ≥90 | 561 (4.76) | 149 (2.46) | 147 (5.45) | 265 (8.74) | | | |
| **Waist circumference(cm)** | 82.59 ± 10.41 | 76.56 ± 8.05 | 85.39 ± 7.24 | 92.16 ± 8.60 | <0.001 | 0 < 1 < 2 | <0.001 |
| M<90, F<85 | 8,558 (72.66) | 5,581 (92.26) | 1,854 (68.72) | 1,123 (37.05) | <0.001 | | <0.001 |
| M≥90, F≥85 | 3,220 (27.34) | 468 (7.74) | 844 (31.28) | 1,908 (62.95) | | | |
| **AST(IU/l)** | 24.51 ± 15.18 | 21.42 ± 13.21 | 23.59 ± 12.40 | 31.50 ± 18.51 | <0.001 | 0 < 1 < 2 | <0.001 |
| **ALT(IU/l)** | 27.43 ± 23.02 | 19.05 ± 13.69 | 26.49 ± 17.88 | 45.03 ± 30.86 | <0.001 | 0 < 1 < 2 | <0.001 |
| **γ-GTP(IU/l)** | 30.83 ± 37.24 | 21.47 ± 27.13 | 32.77 ± 31.23 | 47.82 ± 50.87 | <0.001 | 0 < 1 < 2 | <0.001 |
| **FBS(mg/dl)** | 100.08 ± 18.54 | 95.01 ± 12.89 | 101.74 ± 17.10 | 108.74 ± 24.82 | <0.001 | 0 < 1 < 2 | <0.001 |
| **HbA1c(%)** | 5.62 ± 0.64 | 5.44 ± 0.41 | 5.65 ± 0.63 | 5.94 ± 0.87 | <0.001 | 0 < 1 < 2 | <0.001 |
| <6.5 | 10,968 (93.76) | 5,899 (98.09) | 2,509 (93.58) | 2,560 (85.25) | <0.001 | | <0.001 |
| ≥6.5 | 730 (6.24) | 115 (1.91) | 172 (6.42) | 443 (14.75) | | | |
| **Total cholesterol(mg/dl)** | 205.78 ± 39.22 | 203.42 ± 36.63 | 206.68 ± 39.73 | 209.68 ± 43.24 | <0.001 | 0 < 1 < 2 | <0.001 |
| **Triglyceride(mg/dl)** | 134.09 ± 91.38 | 100.76 ± 55.36 | 147.43 ± 90.96 | 188.89 ± 116.67 | <0.001 | 0 < 1 < 2 | <0.001 |
| **LDL-C(mg/dl)** | 125.06 ± 34.40 | 121.63 ± 32.40 | 127.58 ± 34.90 | 129.69 ± 37.02 | <0.001 | 0 < 1 = 2 | <0.001 |
| **HDL-C(mg/dl)** | 56.50 ± 15.09 | 62.86 ± 15.12 | 52.72 ± 12.62 | 47.14 ± 10.27 | <0.001 | 0 < 1 < 2 | <0.001 |
| **ALP(IU/l)** | 72.76 ± 20.98 | 68.13 ± 20.41 | 76.19 ± 20.42 | 78.98 ± 20.39 | <0.001 | 0 < 1 < 2 | <0.001 |
| **Total bilirubin** | 0.86 ± 0.41 | 0.84 ± 0.41 | 0.86 ± 0.41 | 0.89 ± 0.42 | <0.001 | 0 = 1 < 2 | <0.001 |
| **Direct bilirubin** | 0.31 ± 0.12 | 0.31 ± 0.12 | 0.31 ± 0.12 | 0.32 ± 0.12 | 0.017 | 1 < 2 | 0.044 |
| **Amylase(IU/l)** | 64.19 ± 22.90 | 68.23 ± 23.45 | 62.68 ± 21.48 | 57.45 ± 21.19 | <0.001 | 0 < 1 < 2 | <0.001 |
| <110 | 11,168 (96.68) | 5,656 (95.35) | 2,580 (97.40) | 2,932 (98.69) | <0.001 | | <0.001 |
| ≥110 | 384 (3.32) | 276 (4.65) | 69 (2.60) | 39 (1.31) | | | |
| **Lipase(U/l)** | 31.02 ± 14.35 | 30.61 ± 15.71 | 31.24 ± 12.63 | 31.65 ± 12.82 | 0.009 | 0 < 2 | 0.003 |
| 60 | 9,529 (97.57) | 4,914 (97.97) | 2,189 (97.46) | 2,426 (96.88) | 0.015 | | 0.004 |
| ≥60 | 237 (2.43) | 102 (2.03) | 57 (2.54) | 78 (3.12) | | | |
| **AFP** | 3.34 ± 6.03 | 3.54 ± 8.16 | 3.28 ± 2.48 | 3.01 ± 1.63 | <0.001 | 0 < 2 | <0.001 |
| **hs-CRP(mg/l)** | 0.75 ± 0.83 | 0.52 ± 0.65 | 0.81 ± 0.84 | 1.13 ± 0.98 | <0.001 | 0 < 1 < 2 | <0.001 |
| **Uric acid(mg/dl)** | 5.56 ± 1.50 | 5.03 ± 1.32 | 5.78 ± 1.42 | 6.41 ± 1.48 | <0.001 | 0 < 1 < 2 | <0.001 |

*(Continued)*

**Table 2.** (Continued)

|  | Total | Non-NAFLD | Mild NAFLD | Moderate-to-Severe NAFLD | p-value | Post hoc test | Trend p-value |
|---|---|---|---|---|---|---|---|
| <7 | 9,626 (82.08) | 5,518 (91.59) | 2,135 (79.40) | 1,973 (65.46) | <0.001 |  | <0.001 |
| ≥7 | 2,102 (17.92) | 507 (8.41) | 554 (20.60) | 1,041 (34.54) |  |  |  |
| **BUN(mg/dl)** | 12.60 ± 3.39 | 12.25 ± 3.39 | 12.91 ± 3.47 | 13.01 ± 3.26 | <0.001 | 0 < 1 = 2 | <0.001 |
| **Creatinine(mg/dl)** | 0.85 ± 0.18 | 0.81 ± 0.18 | 0.88 ± 0.18 | 0.91 ± 0.17 | <0.001 | 0 < 1 < 2 | <0.001 |
| **sUA/sCr** | 6.64 ± 1.55 | 6.34 ± 1.43 | 6.71 ± 1.49 | 7.17 ± 1.68 | <0.001 | 0 < 1 < 2 | <0.001 |

Values are presented as means ±SD or number (%).

NAFLD: non-alcoholic fatty liver disease, BMI: body mass index, BP: blood pressure, FBS; Fasting glucose, HbA1C; Hemoglobin A1C, LDL-C: low-density lipoprotein-cholesterol, HDL-C: high-density lipoprotein-cholesterol, AST: aspartate aminotransferase, ALT: alanine aminotransferase, r-GTP: gamma-glutamyl transferase, ALP; Alkaline phosphatase.

**Table 3. Logistic regression analysis (non-NAFLD vs NAFLD).**

|  | NAFLD | p-value |
|---|---|---|
| **Age** | 1.009 (1.003–1.016) | 0.004 |
| **Sex** |  |  |
| Male | 1.660* (1.401–1.966) | <0.001 |
| Female | ref. |  |
| **Smoking** |  |  |
| Non-smoker | ref. |  |
| Ex-Smoker | 1.151 (0.979–1.354) | 0.089 |
| Current-Smoker | 0.958 (0.802–1.144) | 0.634 |
| **Exercise(METs/week)** |  |  |
| <600 | ref. |  |
| ≥600 | 0.956 (0.852–1.072) | 0.438 |
| **BMI(kg/㎡)** |  |  |
| <25 | ref. |  |
| ≥25 | 2.969*(2.594–3.398) | <0.001 |
| **Systolic BP(mmHg)** |  |  |
| <140 | ref. |  |
| ≥140 | 1.115 (0.753–1.652) | 0.586 |
| **Diastolic BP(mmHg)** |  |  |
| <90 | ref. |  |
| ≥90 | 1.339 (0.936–1.915) | 0.110 |
| **Waist circumference(cm)** |  |  |
| M<90, F<85 | ref. |  |
| M≥90, F≥85 | 2.588* (2.200–3.045) | <0.001 |
| **AST(IU/l)** | 0.974* (0.965–0.982) | <0.001 |
| **ALT(IU/l)** | 1.042* (1.035–1.049) | <0.001 |
| **γ-GTP(IU/l)** | 0.999 (0.997–1.001) | 0.521 |
| **FBS(mg/dl)** | 1.021* (1.016–1.027) | <0.001 |
| **HbA1c(%)** |  |  |
| <6.5 | ref. |  |
| ≥6.5 | 1.405 (0.980–2.015) | 0.064 |
| **Total cholesterol(mg/dl)** | 0.992 (0.983–1.000) | 0.052 |

*(Continued)*

**Table 3.** (Continued)

|  | NAFLD | *p*-value |
|---|---|---|
| **Triglyceride(mg/dl)** | 1.007* (1.005–1.009) | <0.001 |
| **LDL-C(mg/dl)** | 1.009 (1.000–1.019) | 0.042 |
| **HDL-C(mg/dl)** | 0.984 (0.975–0.993) | 0.001 |
| **ALP(IU/l)** | 1.005 (1.001–1.008) | 0.003 |
| **Total bilirubin** | 1.003 (0.679–1.481) | 0.990 |
| **Direct bilirubin** | 0.978 (0.256–3.739) | 0.974 |
| **Amylase(IU/l)** |  |  |
| <110 | ref. |  |
| ≥110 | 0.556 (0.394–0.783) | 0.001 |
| **Lipase(U/l)** |  |  |
| 60 | ref. |  |
| ≥60 | 1.160 (0.791–1.703) | 0.447 |
| **AFP** | 0.968 (0.944–0.993) | 0.012 |
| **hs-CRP(mg/l)** | 1.320* (1.221–1.427) | <0.001 |
| **BUN(mg/dl)** | 1.007 (0.990–1.025) | 0.419 |
| **sUA/sCr** | 1.183* (1.137–1.231) | <0.001 |

The adjusted odds ratios for NAFLD of each characteristics with their 95% confidence intervals were estimated by multiple logistic regression analysis. Adjusted for age, sex, smoking, exercise, BMI, sBP, dBP, Waist circumference, AST, ALT, GGT, FBS, HbA1C, Total cholesterol, Triglyceride, LDL.-C, HDL-C, ALP, TB, DB, Amylase, Lipase, AFP, hs-CRP and BUN. Values are presented as OR(95% CI). *Statistically significant values. TB; Total bilirubin, DB; Direct bilirubin, OR; odds ratio, CI; confidence interval.

**Table 4. Logistic regression analysis according to NAFLD severity.**

|  | Mild NAFLD | *p*-value | Moderate-to-Severe NAFLD | *p*-value |
|---|---|---|---|---|
| **Age** | 1.011 (1.004–1.018) | 0.001 | 1.009 (1.000–1.017) | 0.048 |
| **Sex** |  |  |  |  |
| Male | 1.485* (1.238–1.780) | <0.001 | 2.346* (1.869–2.946) | <0.001 |
| Female | ref. |  | ref. |  |
| **Smoking** |  |  |  |  |
| Non-smoker | ref. |  | ref. |  |
| Ex-Smoker | 1.170 (0.983–1.393) | 0.077 | 1.125 (0.918–1.379) | 0.256 |
| Current-Smoker | 0.971 (0.802–1.176) | 0.767 | 0.910 (0.730–1.134) | 0.401 |
| **Exercise(METs/week)** |  |  |  |  |
| <600 | ref. |  | ref. |  |
| ≥600 | 1.007 (0.891–1.138) | 0.908 | 0.846 (0.729–0.981) | 0.027 |
| **BMI(kg/㎡)** |  |  |  |  |
| <25 | ref. |  | ref. |  |
| ≥25 | 2.442* (2.109–2.827) | <0.001 | 4.549* (3.810–5.431) | <0.001 |
| **Systolic BP(mmHg)** |  |  |  |  |
| <140 | ref. |  | ref. |  |
| ≥140 | 1.142 (0.751–1.736) | 0.536 | 1.079 (0.676–1.724) | 0.749 |
| **Diastolic BP(mmHg)** |  |  |  |  |
| <90 | ref. |  | ref. |  |
| ≥90 | 1.180 (0.802–1.735) | 0.400 | 1.611 (1.058–2.451) | 0.026 |

(*Continued*)

**Table 4.** (Continued)

| | Mild NAFLD | *p*-value | Moderate-to-Severe NAFLD | *p*-value |
|---|---|---|---|---|
| **Waist circumference(cm)** | | | | |
| M<90, F<85 | ref. | | ref. | |
| M≥90, F≥85 | 1.927* (1.614–2.301) | <0.001 | 3.825* (3.166–4.622) | <0.001 |
| **AST(IU/l)** | 0.976* (0.966–0.986) | <0.001 | 0.970* (0.960–0.980) | <0.001 |
| **ALT(IU/l)** | 1.028* (1.021–1.036) | <0.001 | 1.060* (1.052–1.068) | <0.001 |
| **γ-GTP(IU/l)** | 1.000 (0.997–1.002) | 0.718 | 0.999 (0.997–1.002) | 0.666 |
| **FBS(mg/dl)** | 1.019* (1.014–1.025) | <0.001 | 1.027* (1.021–1.033) | <0.001 |
| **HbA1c(%)** | | | | |
| <6.5 | ref. | | ref. | |
| ≥6.5 | 1.175 (0.796–1.734) | 0.417 | 1.769 (1.165–2.686) | 0.007 |
| **Total cholesterol(mg/dl)** | 0.992 (0.983–1.001) | 0.094 | 0.989 (0.979–1.000) | 0.050 |
| **Triglyceride(mg/dl)** | 1.006* (1.005–1.008) | <0.001 | 1.008* (1.006–1.010) | <0.001 |
| **LDL-C(mg/dl)** | 1.009 (0.999–1.018) | 0.073 | 1.012 (1.000–1.023) | 0.043 |
| **HDL-C(mg/dl)** | 0.987 (0.977–0.997) | 0.009 | 0.976* (0.964–0.988) | <0.001 |
| **ALP(IU/l)** | 1.006* (1.003–1.009) | <0.001 | 1.002 (0.998–1.006) | 0.356 |
| **Total bilirubin** | 1.026 (0.677–1.554) | 0.904 | 0.956 (0.580–1.576) | 0.859 |
| **Direct bilirubin** | 0.904 (0.217–3.772) | 0.890 | 1.036 (0.184–5.836) | 0.968 |
| **Amylase(IU/l)** | | | | |
| v<110 | ref. | | ref. | |
| ≥110 | 0.621 (0.434–0.887) | 0.009 | 0.395 (0.232–0.672) | 0.001 |
| **Lipase(U/l)** | | | | |
| 60 | ref. | | ref. | |
| ≥60 | 1.125 (0.748–1.692) | 0.572 | 1.271 (0.794–2.034) | 0.318 |
| **AFP** | 0.984 (0.963–1.005) | 0.133 | 0.916* (0.879–0.954) | <0.001 |
| **hs-CRP(mg/l)** | 1.255* (1.155–1.364) | <0.001 | 1.456* (1.327–1.598) | <0.001 |
| **BUN(mg/dl)** | 1.008 (0.989–1.027) | 0.416 | 1.005 (0.982–1.028) | 0.678 |
| **sUA/sCr** | 1.147* (1.099–1.196) | <0.001 | 1.275* (1.212–1.341) | <0.001 |

The adjusted odds ratios for NAFLD of each characteristics with their 95% confidence intervals were estimated by multiple logistic regression analysis. Adjusted for age, sex, smoking, exercise, BMI, sBP, dBP, Waist circumference, AST, ALT, GGT, FBS, HbA1C, Total cholesterol, Triglyceride, LDL.-C, HDL-C, ALP, TB, DB, Amylase, Lipase, AFP, hs-CRP and BUN. Values are presented as OR(95% CI). *Statistically significant values. TB; Total bilirubin, DB; Direct bilirubin, OR; odds ratio, CI; confidence interval.

levels. Several recent studies have revealed a relationship between sUA levels and dyslipidemia [22], metabolic syndrome [33], hypertension, diabetes, and chronic kidney disease [34, 35] based on insulin resistance. Insulin resistance is also associated with cardiovascular diseases [10]; therefore, its management is essential. Additionally, sUA levels are associated with NAFLD [13–17], and sUA to HDL-C ratio (UHR), a uric acid-based marker, is also associated

**Table 5. sUA/sCr AUC, cut-off value, sensitivity, specificity.**

| sUA/sCr | AUC (95% CI) | Cut-off value | Sensitivity (%) | Specificity (%) | *p*-value |
|---|---|---|---|---|---|
| **Non-NAFLD** | 0.619 (0.610–0.629) | 6.58 | 57.9 | 61.4 | <0.001 |
| **Mild NAFLD** | 0.581 (0.566–0.595) | 6.43 | 56.7 | 57.3 | <0.001 |
| **Moderate-to–Severe NAFLD** | 0.653 (0.642–0.666) | 6.73 | 59.0 | 65.2 | <0.001 |

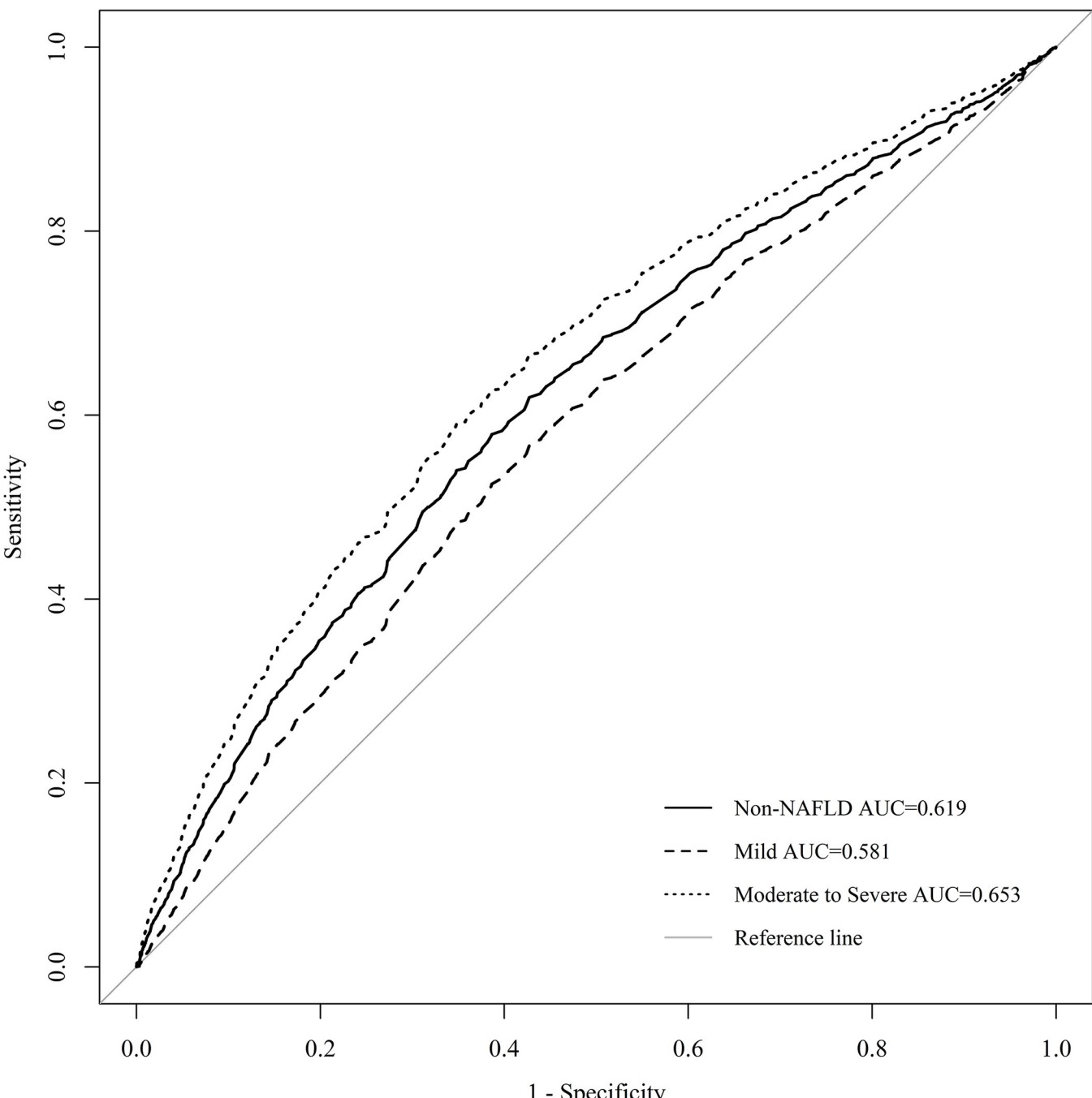

**Fig 2. sUA/sCr ROC curve for identifying NAFLD.** The diagnostic accuracy of the sUA/sCr ratio in separating participants with NAFLD severity was analyzed by using the ROC method.

with NAFLD [36]. UA induces mitochondrial oxidant production via the purine decomposition pathway [11]. Endoplasmic reticulum (ER) stress caused by UA leads to increased GRP78/94 expression, splicing of XBP-1, and phosphorylation of PERK and eIF-2α to activate SREBP-1c, which promotes liver fat production through overexpression of ACC1, FAS, and SCD1. In addition, UA-induced NADPH oxidase activation before ER stress induces mitochondrial reactive oxygen species production in hepatocytes [12]. Mitochondrial oxidative

**Table 6. sUA AUC, cut-off value, sensitivity, specificity.**

| sUA | AUC (95% CI) | Cut-off value | Sensitivity (%) | Specificity (%) | *p*-value |
|---|---|---|---|---|---|
| Non-NAFLD | 0.710 (0.704–0.719) | 5.25 | 70.9 | 60.6 | <0.001 |
| Mild NAFLD | 0.655 (0.642–0.667) | 5.25 | 62.7 | 60.6 | <0.001 |
| Moderate-to–Severe NAFLD | 0.760 (0.751–0.769) | 5.35 | 76.0 | 63.2 | <0.001 |

stress inhibits the aconitase of the Krebs cycle, leading to citrate accumulation and stimulation of ATP citrate lyase and fat-acid synthase; this leads to de novo lipogenesis [11], causing fatty liver. sUA, the final metabolic product of purines, is excreted through the kidneys. Therefore, increased UA levels due to kidney failure caused by hypertension or diabetes make it difficult to accurately reflect endogenous UA levels [37]. Therefore, the sUA/sCr ratio, the UA level corrected for kidney function, has emerged [18–22].

In a previous study, Seo and Han [20] revealed a post-correction OR of sUA/sCr in NAFLD as 0.182 (95% CI: 1.066–1.311, $p = 0.002$), while Ma et al. [22] reported a post-correction OR of sUA/sCr in NAFLD as 1.529 (95% CI: 1.011–2.310, $p = 0.044$). Wang et al. [21] reported that there was a nonlinear relationship between sUA/sCR and NAFLD risk and that when sUA/sCR was less than 4.425, the OR was 1.551 (95% CI: 1.504–1.599) and the AUC was 0.624 (95% CI: 0.6589–0.6660) with 64.8% sensitivity and 59.8% specificity. However, these studies had limitations in that ultrasound, which is the most widely used modality for fatty liver diagnosis today, was not used or targeted for Koreans.

In this study, a positive relationship was identified between NAFLD and sUA/sCr, and when sUA/sCr increased by 1, the risk in the mild NAFLD group increased by 1.147 times (OR = 1.147, 95% CI: 1.099–1.196; $p < 0.001$) and the risk of moderate-to-severe NAFLD group increased by 1.275 times (OR = 1.275, 95% CI:1.212–1.341, $p < 0.001$). The AUC, sensitivity, and specificity of sUA/sCr according to the severity of NAFLD were 0.619 (95% CI: 0.610–0.629), 57.9%, and 61.4% ($p < 0.001$) in the presence of NAFLD; 0.581 (95% CI: 0.566–0.595), 56.7%, and 57.3% for mild NAFLD; and 0.653 (95% CI: 0642–0.666), 59.0%, and 65.2% ($p < 0.001$) for moderate-to-severe NAFLD. Therefore, our results are consistent with those of previous studies [21–23].

This study has several limitations. First, the study participants visited a particular university hospital for medical examinations; thus, they were not representative of the entire general population. Second, because this was a cross-sectional study, an exact causal relationship could not be identified. Third, since information on drug use, medical history, alcohol consumption, smoking history, and exercise was completely dependent on the self-administered survey, uncontrolled or unconfirmed confounding factors may still exist. Fourth, although the moderate-to-severe fatty liver disease group included 2,000 moderate and severe NAFLD patients and 19 with liver cirrhosis, the number of patients in this group was smaller than that in the other groups, making accurate comparisons difficult. Finally, because NAFLD was diagnosed using ultrasonography rather than liver biopsy, the accuracy of the diagnosis may have decreased. However, abdominal ultrasonography is already widely used in epidemiological studies for the diagnosis of NAFLD [38] and is currently widely used to confirm the degree of intrahepatic adiposity [39].

## Conclusion

In this study, a positive relationship between the sUA/sCr ratio and the severity of NAFLD diagnosed using ultrasound was confirmed. As there have been no previous studies comparing the relationship between sUA/sCR and the severity of NAFLD, the results of our study can be

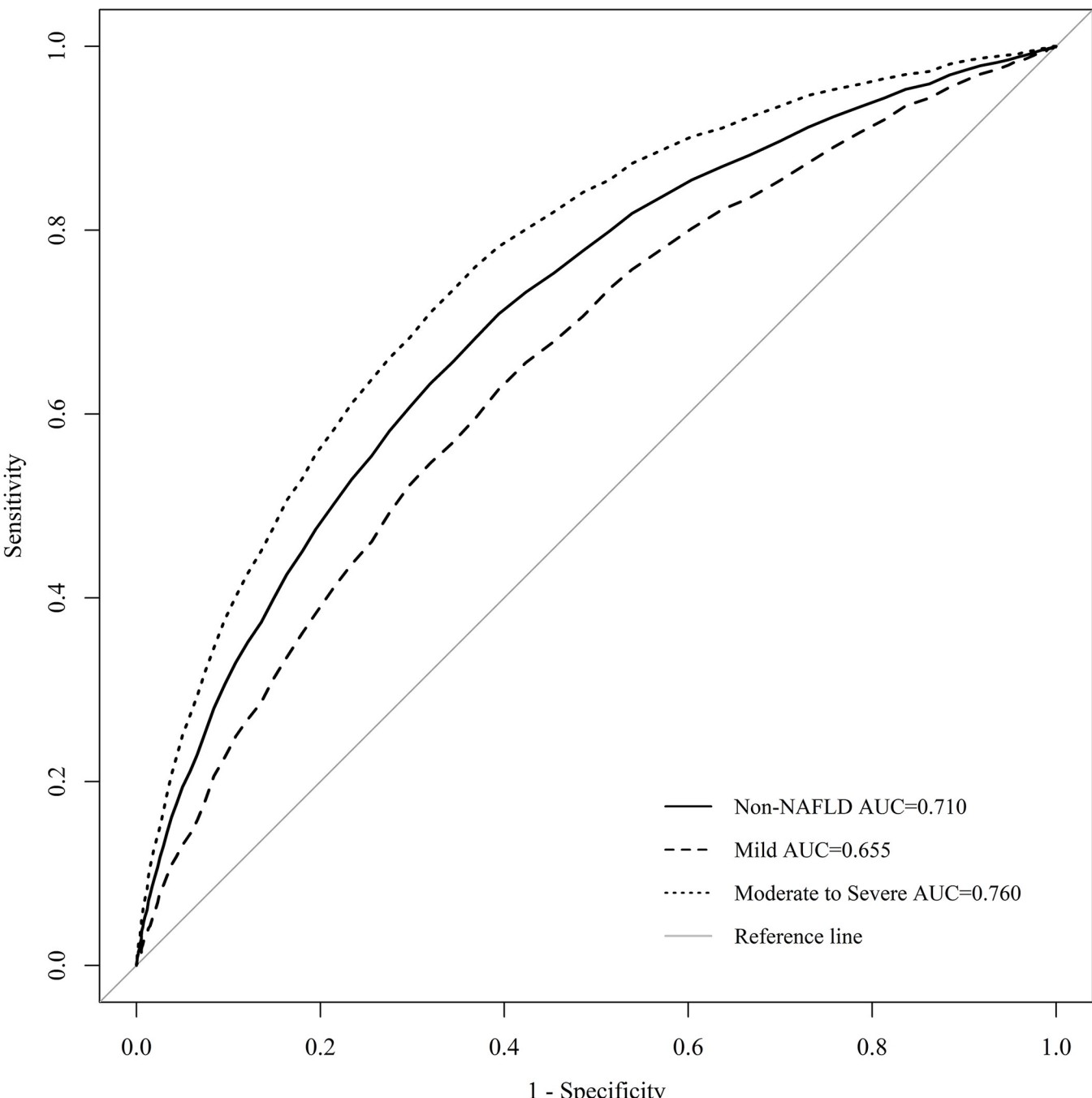

**Fig 3. sUA ROC curve for identifying NAFLD.** The diagnostic accuracy of the serum Uric acid in separating participants with NAFLD severity was analyzed by using the ROC method.

useful for real-world practice because the ultrasound results were analyzed with a large number of Korean participants. In addition, through this study, it was confirmed that the sensitivity and specificity of sUA/sCR for NAFLD were lower than those of sUA. This seems to explain why the sUA/sCR ratio has not yet been widely used as a diagnostic indicator for NAFLD. Follow-up studies to confirm the relationship between the sUA/sCr ratio and sUA/sCr with non-

alcoholic steatohepatitis or liver cirrhosis, and to confirm the causal relationship through prospective studies are necessary.

## Acknowledgments

We would like to thank Editage (www.editage.co.kr) for English language editing.

## Author Contributions

**Conceptualization:** Jangwon Choi, Hyun Joe.

**Data curation:** Jangwon Choi, Hwang-Sik Shin, Nam Hun Heo.

**Formal analysis:** Nam Hun Heo.

**Investigation:** Jangwon Choi, Yong-Jin Cho.

**Methodology:** Jangwon Choi, Jung-Eun Oh.

**Project administration:** Hyun Joe.

**Resources:** Yong-Jin Cho, Hwang-Sik Shin.

**Software:** Hwang-Sik Shin.

**Supervision:** Hyun Joe.

**Validation:** Hyun Joe, Hwang-Sik Shin.

**Writing – original draft:** Jangwon Choi.

**Writing – review & editing:** Jangwon Choi.

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
