## [Decision Letter · Decision Letter 0]

12 Jun 2023

PONE-D-23-16163The Correlation Between NAFLD and Serum Uric Acid to Serum Creatinine RatioPLOS ONE

Dear Dr. Joe,

Thank you for submitting your manuscript to PLOS ONE. After careful consideration, we feel that it has merit but does not fully meet PLOS ONE’s publication criteria as it currently stands. Therefore, we invite you to submit a revised version of the manuscript that addresses the points raised during the review process.

ACADEMIC EDITOR: Statistics must be elaborated. According to the data presented in statistics section, authors stated that 'Student’s t-test, or one-way analysis of variance' was used in comparison of the variables. These tests are used in comparison of the variables that fit into normal distribution. Which test was conducted for normality analysis? Please state appropriately. 

We look forward to receiving your revised manuscript.

Kind regards,

Gulali Aktas

Academic Editor

PLOS ONE

Journal Requirements:

4. Please clarify the number of Figures/Tables uploaded in your manuscript and PDF file. 

5. Please include a caption for figure 3.

6. Please include a copy of Tables 5 and 6 which you refer to in your text on page 21.

Additional Editor Comments:

Statistics must be elaborated. According to the data presented in statistics section, authors stated that 'Student’s t-test, or one-way analysis of variance' was used in comparison of the variables. These tests are used in comparison of the variables that fit into normal distribution. Which test was conducted for normality analysis? Please state appropriately.

Reviewers' comments:

Reviewer's Responses to Questions

**Comments to the Author**

1. Is the manuscript technically sound, and do the data support the conclusions?

Reviewer #1: Yes

Reviewer #2: Yes

2. Has the statistical analysis been performed appropriately and rigorously? 

Reviewer #1: Yes

Reviewer #2: Yes

3. Have the authors made all data underlying the findings in their manuscript fully available?

Reviewer #1: Yes

Reviewer #2: Yes

4. Is the manuscript presented in an intelligible fashion and written in standard English?

Reviewer #1: Yes

Reviewer #2: Yes

5. Review Comments to the Author

Reviewer #1: Dear Authors

Thank you very much for the opportunity of reviewing this manuscript for kournal. The article is about positive relationship between the sUA/sCr ratio and the severity of NAFLD. Indeed, the article is very interesting and of utmost importance to the field.

Title is appropriate for article, summary abstracted the manuscript adequately, the keyword is enough. The background and the rationale of the study sufficiently expressed in introduction. Study cohort, laboratory and statistical analyses are well issued in methodology. Results of the study are interesting and given in an easy to read way. Discussion is faultless and is the most perfect section of the manuscript. It does not require any more revisions. References are accurate. 12. The quality of the figures should be improved. Table is informative. As a result, the article should be accepted for publication in the journal.

Reviewer #2: Authors should improve the discussion. Serum uric acid and uric acid based markers are reported to be associated with inflammatory conditions such as hypertension (Postgraduate Medicine 2022;134(3): 297-302. DOI: 10.1080/00325481.2022.2039007), thyroiditis (Rom. J. Intern. Med. 2021;59(4):403-408. DOI: 10.2478/rjim-2021-0023.), metabolic syndrome (Rev Assoc Med Bras 2019; 65(1):9-15. DOI: 10.1590/1806-9282.65.1.9) type 2 DM (Aging Male. 2020;23(5):1098-1102. DOI: 10.1080/13685538.2019.1678126), and diabetic kidney disease (Postgraduate Medicine, 2023. DOI: 10.1080/00325481.2023.2214058). Moreover,the ABUND study showed that uric acid based markers are associated with non-alcoholic fatty liver disease (Rev Assoc Med Bras 2021;67(4):549-554. DOI: 10.1590/1806-9282.20201005.).

6. PLOS authors have the option to publish the peer review history of their article (what does this mean?). If published, this will include your full peer review and any attached files.

Reviewer #1: No

Reviewer #2: No

---

## [Author Response · Author response to Decision Letter 0]

15 Jun 2023

We, the authors, appreciate the thoughtful and detailed reviews of the reviewers.

Normality tests were conducted on continuous variables using both the Kolmogorov-Smirnov test and the Shapiro-Wilk test. Student’s t-test, or one-way analysis of variance, was used to analyze continuous variables, whereas Pearson's chi-square test was used to analyze categorical variables. This correction was reflected in the manuscript.(page 9, 153-156)

In response to reviewer #2's comment, I have added previous findings(page 22, 295-296) and reference(37).

The authors have not received specific funding for this work so far. After this manuscript is published, we are planning to apply the research fund program of the Soonchunhyang University Industry-University Cooperation Foundation. This content was corrected in the manuscript.

The figure and table numbering has been corrected.

And all figures are revised.

All relevant data are within the manuscript.

*** No Supporting Information files.

Journal Requirements:

#1. Ensure that your manuscript meets PLOS ONE's style requirements: Confirmed

#2. financial disclosure: Confirmed

#3. Data Availability statement: All relevant data are within the manuscript.

#4. Clarify the number of Figures/Tables uploaded in your manuscript and PDF file.: Confirmed

#5. Include a caption for figure 3.: Confirmed

#6. Includ a copy of Tables 5 and 6 which you refer to in your text on page 21.: Confirmed

#7. Review your reference list to ensure that it is complete and correct.: Confirmed

I look forward to your positive reply.

---

## [Decision Letter · Decision Letter 1]

2 Jul 2023

The Correlation Between NAFLD and Serum Uric Acid to Serum Creatinine Ratio

PONE-D-23-16163R1

Dear Dr. Joe,

We’re pleased to inform you that your manuscript has been judged scientifically suitable for publication and will be formally accepted for publication once it meets all outstanding technical requirements.

Kind regards,

Gulali Aktas

Academic Editor

PLOS ONE

Additional Editor Comments (optional):

Reviewers' comments:

Reviewer's Responses to Questions

**Comments to the Author**

1. If the authors have adequately addressed your comments raised in a previous round of review and you feel that this manuscript is now acceptable for publication, you may indicate that here to bypass the “Comments to the Author” section, enter your conflict of interest statement in the “Confidential to Editor” section, and submit your "Accept" recommendation.

Reviewer #1: All comments have been addressed

Reviewer #2: All comments have been addressed

2. Is the manuscript technically sound, and do the data support the conclusions?

Reviewer #1: Yes

Reviewer #2: Yes

3. Has the statistical analysis been performed appropriately and rigorously? 

Reviewer #1: Yes

Reviewer #2: Yes

4. Have the authors made all data underlying the findings in their manuscript fully available?

Reviewer #1: Yes

Reviewer #2: Yes

5. Is the manuscript presented in an intelligible fashion and written in standard English?

Reviewer #1: Yes

Reviewer #2: Yes

6. Review Comments to the Author

Reviewer #1: De ar authors

The manuscript should be accepted for publication for your journal

Reviewer #2: The authors addressed all of my comments in the previous review round. I think the paper does not require further revision. Hence,i I suggest publication in its current form.

7. PLOS authors have the option to publish the peer review history of their article (what does this mean?). If published, this will include your full peer review and any attached files.

Reviewer #1: No

Reviewer #2: No

---

## [Editor Report · Acceptance letter]

11 Jul 2023

PONE-D-23-16163R1 

The Correlation Between NAFLD and Serum Uric Acid to Serum Creatinine Ratio 

Dear Dr. Joe:

I'm pleased to inform you that your manuscript has been deemed suitable for publication in PLOS ONE. Congratulations! Your manuscript is now with our production department. 

Kind regards, 

on behalf of

Professor Gulali Aktas 

Academic Editor

PLOS ONE